# Cetylpyridinium Chloride-Containing Mouthwashes Show Virucidal Activity against Herpes Simplex Virus Type 1

**DOI:** 10.3390/v15071433

**Published:** 2023-06-25

**Authors:** Eva Riveira-Muñoz, Edurne Garcia-Vidal, Manuel Bañó-Polo, Rubén León, Vanessa Blanc, Bonaventura Clotet, Ester Ballana

**Affiliations:** 1IrsiCaixa AIDS Research Institute, Health Research Institute Germans Trias i Pujol (IGTP), Hospital Germans Trias i Pujol, Universitat Autònoma de Barcelona, 08916 Badalona, Spain; eriveira@irsicaixa.es (E.R.-M.); egvidal@irsicaixa.es (E.G.-V.); bclotet@irsicaixa.es (B.C.); 2Department of Microbiology, Dentaid Research Center, 08290 Cerdanyola del Vallès, Spain; manuel.bano@dentaid.es (M.B.-P.); leon@dentaid.es (R.L.); blanc@dentaid.es (V.B.)

**Keywords:** HSV, HPV, CPC, oral infection, enveloped viruses

## Abstract

The oral cavity is particularly susceptible to viral infections that are self-recovering in most cases. However, complications may appear in severe cases and/or immunocompromised subjects. Cetylpyridinium chloride (CPC)-containing mouthwashes are able to decrease the infectivity of the SARS-CoV-2 virus by disrupting the integrity of the viral envelope. Here, we show that CPC, as the active ingredient contained in commercialized, exerts significant antiviral activity against enveloped viruses, such as HSV-1, but not against non-enveloped viruses, such as HPV. CPC-containing mouthwashes have been used as antiseptics for decades, and thus, they can represent a cost-effective measure to limit infection and spread of enveloped viruses infecting the oral cavity, aiding in reducing viral transmission.

## 1. Introduction

The oral cavity is especially susceptible to viral infections because of its anatomy, particularly its soft tissue and salivary glands [1,2]. Several viruses are associated with oral disease-causing primary lesions, but also, oral mucosa can be affected by the secondary pathological processes of bacterial or fungal nature due to viral immunosuppression, such as those associated with the human immunodeficiency virus infection (HIV) [3,4,5]. 

Although many different viruses can affect the oral cavity, members of the herpesvirus and human papillomavirus (HPV) families cause the most common primary oral viral infections [1]. HSVs contain a double-stranded linear DNA molecule enveloped by an icosahedral capsid and a lipid envelope [6]. Two distinct types of HSV have been identified as human pathogens, HSV-1 and HSV-2, and although both viruses have been associated with oral diseases, only HSV-1 targets the oral cavity [7]. HPVs, on the other hand, are non-enveloped viruses containing double-stranded DNA [8]. Over 100 subtypes of HPV have been identified, with at least 13 correlated to the appearance of oral lesions [9]. Despite their high prevalence, most viral infections in the oral cavity have a subclinical course, with clinical manifestations limited to oral lesions that resolve without intervention and are easily managed with supportive therapy. However, leaving severe conditions untreated may lead to significant comorbidity, especially in already immunocompromised individuals [10]. For example, certain viruses are implicated in the development of dysplastic and neoplastic transformations of squamous epithelium [11]. 

Cetylpyridinium chloride (CPC) is a cationic ammonium compound with surfactant properties that is safe to be used in humans in a wide range of concentrations. It is often found in products at concentrations ranging from 0.05 to 0.1% (0.5–1 mg/mL), such as mouthwashes, toothpastes, oral tablets, deodorants, and aphthae-treating products, and its use is indicated as an anti-bacterial agent [12,13,14]. CPC can disrupt the lipid membrane through physicochemical interactions, and compared to other ingredients in mouthwashes, including povidone-iodine and chlorhexidine (CHX), CPC is tasteless and odorless and is, therefore, suitable for application via oral care products. Mouth rinses have traditionally been focused on prevention or used as adjuvants in the treatment of periodontal diseases [3,12,15]. In fact, in addition to its well-reported bactericidal effects, CPC has been reported to have antiviral effects against the influenza virus [16] and, even more recently, against coronaviruses [1,17,18,19,20]. Indeed, during the COVID-19 pandemic, CPC was tested for its ability to reduce SARS-CoV-2 virions in the oral cavity, confirmed in vitro but also in clinical trials [21,22,23,24,25]. Here, we aim to test the use of commercial mouthwashes containing CPC as anti-virucidal agents against viruses causing oral cavity infections to provide additional measures to limit infection and spread that might aid in reducing viral transmission. 

## 2. Materials and Methods

### 2.1. CPC-Containing Mouthwashes Employed 

Two CPC-containing formulations from Dentaid SL with different intended uses were included, Vitis CPC Protect and PerioAid Intensive Care (Table 1). Both mouthwashes are commercially available, and the composition of the vehicles used in this study is as follows: deionized water, humectant, sweetener, emulsifier, aroma, and preservatives. In addition, PerioAid Intensive Care (0.05% CPC) contains 0.12% chlorhexidine and is intended for limited-term use as a coadjuvant by patients undergoing periodontal treatment and after surgery in the oral cavity. Vitis CPC Protect (0.07% CPC) is recommended as a daily-use product to prevent and reduce dental plaque formation. 

### 2.2. Cell Cultures and Viral Stocks 

Vero E6 cells (ATCC CRL-1586) and 293TT (ATCC CRL-3467) cells were maintained in Dulbecco’s modified Eagle medium (DMEM; Invitrogen, Waltham, MA, USA), supplemented with 10% fetal bovine serum (Invitrogen), 100 U/mL penicillin, and 100 µg/mL streptomycin (all ThermoFisher Scientific, Waltham, MA, USA).

HSV-1 was grown in Vero cells and stored at −80 °C until use. HSV-1 virus is a laboratory-adapted 17syn+ strain containing a cytomegalovirus (CMV)-green fluorescent protein (GFP) cassette in the US5 region [26].

L1-L2 HPV 16 pseudovirus was generated by co-transfection in 293TT cells following the Standard Production Protocol from the Center for Cancer Research in Lab of Cellular Oncology Technical Files (https://home.ccr.cancer.gov/lco/production.asp, accessed on 25 May 2022). In brief, two plasmids, p16SheLL containing HPV16 L1 + L2 genes and pfwB reporter plasmid containing GFP (both from Addgene, Watertown, MA, USA), were transfected in 293TT cells plated the day before at a 40% confluence, using lipofectamine (Thermofisher). Culture supernatant and cells were collected 48 h post-transfection, and after centrifuging and washing, cells were lysed to release produced pseudovirus, and viral stock was frozen at −80 °C until used. p16sheLL and pfwB were gifts from John Schiller (Addgene plasmids #37320 and #37329).

Both viral stocks were titrated in Vero E6 cells prior to the antiviral activity test. Briefly, Vero cells were plated and challenged in triplicate with sequential 1/5 dilutions of the viral stocks, and infection was measured 48 h later as a percentage of GFP-positive cells by flow cytometry (LSRII; BD Bioscience, Madrid, Spain)

### 2.3. Antiviral Activity against HSV and HPV 

To mimic the procedure of mouthwashes, 1 × 10^4^ TCID_50_/_mL_ of corresponding HSV or HPV viruses were incubated with mouthwash-based solutions or PBS for 2 min at RT. Then, incubated viral solutions were washed 4 times by adding 10 mL of PBS and centrifugation with Macrosep Advance 100K Centrifugal Devices (Pall Corporation, Madrid, Spain), 2500× *g* for 14 min. Viruses were recovered from the upper membrane of the centrifuge tubes and immediately used for antiviral determination by adding 100 μL of the recovered virus to 1.5 × 10^4^ Vero E6 cells plated in 96-well plates. Antiviral activity was measured 48 h after infection as a percentage of GFP-expressing cells relative to untreated control by flow cytometry (LSRII; BD Biosciences, Madrid, Spain). Cell viability was measured by flow cytometry using Live/Dead^TM^ Fixable Near-IR Dead Cell Stain kit (Cat. num L34975, ThermoFisher Scientific, Waltham, MA, USA), together with absolute quantification of live cells according to forward scatter and side scatter parameters. Data were analyzed using FlowJo software (BD Biosciences, Madrid, Spain).

### 2.4. Statistical Analysis

Experimental data were analyzed with the PRISM statistical package. Data are expressed as mean ± SD of four independent experiments. *p*-values were calculated using an unpaired, two-tailed, *t*-student test.

## 3. Results and Discussion

Given the reported effect of CPC, we sought to assess the potential antiviral properties of CPC-containing mouthwashes against HSV-1 and HPV infection in epithelial cells, with the final aim of providing additional proof of the efficacy of CPC-containing mouthwashes in reducing viral load in the oral cavity [27,28,29].

To explore the suitability of CPC-containing mouthwash solutions as antivirals, several formulations for oral care were prepared in the chemical research laboratory of the DENTAID Research Center. Two distinct CPC-containing formulas were evaluated: Vitis CPC Protect, containing 0.07% CPC, an oral rinse recommended for daily use to prevent and reduce dental plaque formation, and PerioAid Intensive Care, containing 0.05% CPC together with 0.12% chlorhexidine, intended for limited-term use as a coadjuvant for patients undergoing periodontal treatment and/or surgery in the oral cavity. Control formulations included PerioAid and Vitis vehicles without active principles, 0.05% CPC in PBS, 2% SDS in PBS, and also a PerioAid-based formulation containing CPC or chlorhexidine alone (Figure 1, Table 1). 

First, we determined the toxicity of CPC formulations on cells that are susceptible to viral infection due to the high cytotoxicity reported for CPC in the literature [26], a consequence of its strong affinity for the phospholipid bilayer of the plasma membrane. The different formulations were tested on Vero cells, epithelial cells commonly used for antiviral testing [26], following an in vitro protocol intended to mimic the procedure of a mouthwash, i.e., short-time exposure (1 or 2 min) followed by extensive washing. Cell viability was measured by flow cytometry, using a specific dye for detecting dead cells (Figure 2A). As shown in Figure 2A,B, none of the tested formulations significantly compromised cell viability, except for CPC alone (mean viability = 71.9% ± 17.2; *p*-value = 0.0742, paired *t*-test) and PerioAid (mean viability = 83.3% ± 4.1; *p*-value = 0.059; paired *t*-test) although viable cells were above 70% in most of the cases. These results indicate the suitability of the protocol used to test the antiviral activity of the formulations, in accordance with the wide use of CPC as an antiseptic and its acceptance by both the FDA (UNII D9OM4SK49P) and the Scientific Committee on Consumer Safety of European Union (SCCS/1548/15) for its use in gingivitis as a mouthwash as long as CPC concentration does not exceed 0.1% (*w/w*). 

Subsequently, the potential antiviral role of CPC-containing mouthwash against HSV and HPV was assessed in the same experimental setting. HSV and HPV viral stocks were pretreated for 2 min with the different formulations and also with SDS as a positive control (Figure 2C–F). SDS is a broad-spectrum surfactant [29] that has been proven effective as a topical microbicide and viral inactivator against enveloped and non-enveloped viruses, causing dissociation of the viral envelope and capsid proteins, including those of HIV (human immunodeficiency virus), HPV (human papillomavirus) and HSV (herpes simplex virus) [29]. As expected, SDS was able to completely block infection by both viruses (Figure 2D,F, third bar). The same results were observed for CPC-containing mouthwashes in the case of the enveloped HSV virus (Figure 2C,D), whereas no significant inhibition was seen in HPV (Figure 2E,F). The vehicle formulations of both mouthwashes tested showed no effect in any of the cases, indicating that CPC is the active principle responsible for virucidal activity. 

Overall, our results demonstrate that CPC-containing mouthwashes may represent an easy, affordable, safe, and feasible approach for the prevention of infections by enveloped viruses, such as highly prevalent HSV infections. Oral transmission of oral viral infections is mainly through contact with the virus in sores, saliva, or surfaces in or around the mouth. Therefore, limiting viral load in the oral cavity may represent an interesting approach not only for limiting reinfections to prevent complications in cases of severe infection but also for reducing viral transmission. Indeed, poor oral hygiene with plaque build-up, followed by the onset of gingivitis and periodontitis, also facilitates direct entry of viruses via the gingival sulci and periodontal pockets, facilitating the spread of infection to other tissues [9]. Thus, the anti-plaque effect of CPC in mouthwashes may synergize with its virucidal activity to lessen the risk of more severe infections. 

Previous reports have already demonstrated the antiviral activity of CPC against different variants of SARS-CoV-2 [1,17,18,19,20] and the influenza virus [16], exerting its activity by inhibiting viral fusion on target cells due to CPC’s capacity to disrupt the integrity of the viral envelope. In contrast, it was suggested that CPC blocks HSV replication by interfering with the translocation of NF-κB into the nucleus of HSV-infected cells [30,31]. However, although possible, our results point towards the envelope disruption hypothesis, as in our experimental setting, it was the viral stocks and not the cells that were exposed to CPC prior to extensive washing, and only enveloped viruses were effectively blocked by short-time exposures. Based on these results, we hypothesize that the effect of CPC observed on enveloped viruses (coronaviruses or HSV) would be meager on non-enveloped viruses (HPV), as protein capsids are not disturbed by quaternary ammonium compounds.

One of the limitations of our study is the lack of in vivo data and further analysis of the infectivity of live viruses in clinical samples. However, previous data from COVID-19 patients indicated that virus inactivation with oral preparations containing above 0.05% of CPC for at least 20 s is considered to be effective in limiting SARS-CoV-2 viral load. Here, in an attempt to mimic the recommended procedure of mouthwashes, we also performed short-term incubations of viral stocks with CPC-containing mouthwashes, also suggesting that their use might also be useful against other viruses infecting the oral cavity but limited to those that are enveloped. Moreover, despite the reported temporary effects of CPC-containing mouthwashes, they undoubtedly reduce viral load effectively, suggesting that habitual use at regular intervals may reduce the risk of viral transmission. Further studies in patients are needed to validate that continuous use of CPC formulations could be considered an effective risk-mitigation strategy against viral transmission and infection of enveloped viruses. 

On the whole, these findings suggest that formulations containing CPC are effective in limiting enveloped virus replication. CPC, which is one of the most commonly used antiseptics in oral care products, has been confirmed to be safe for oral use, with fewer reported side effects than povidone-iodine and chlorhexidine, which are used as active ingredients in oral care products. By contrast, extensive studies demonstrated the safety of CPC for human use; therefore, the CPC is considered to be suitable for routine use compared to other antiviral agents. Given the experience with the COVID-19 global pandemic, and the recognition of the significance of the oral cavity in infection, transmission, and disease severity, daily use of an effective CPC mouthwash as part of a good oral hygiene routine could be a low-cost and simple measure to reduce enveloped-virus transmission risk. 

## Figures and Tables

**Figure 1 viruses-15-01433-f001:**
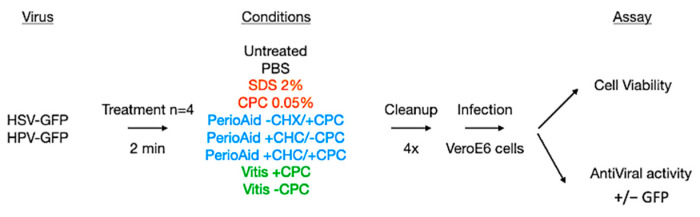
**In vitro evaluation of CPC-containing mouthwash antiviral activity.** Schematic representation of the workflow used for evaluating mouthwashes against HPV and HSV viruses.

**Figure 2 viruses-15-01433-f002:**
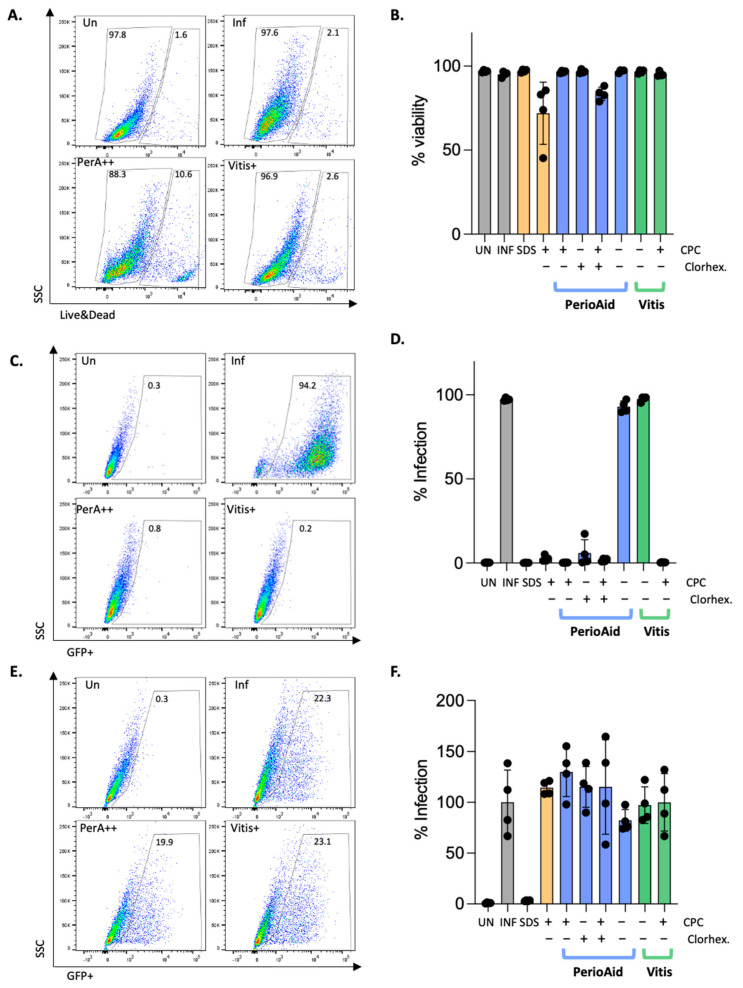
**In vitro evaluation of CPC**-**containing mouthwash antiviral activity.** (**A**) Flow cytometry plots of Live/Dead staining of cells treated with the corresponding conditions. (**B**) Viability of cells treated in vitro with CPC-containing mouthwash protocol. Data represents % of viable cells in each condition. (**C**) Flow cytometry plots of HSV in vitro infection, measured by GFP expression. (**D**) Blockade of HSV infection after pretreatment with the different CPC formulations tested. Bars represent the percentage of GFP-expressing cells measured by flow cytometry. (**E**) Flow cytometry plots of HPV in vitro infection, measured by GFP expression. (**F**) Relative HPV infection after pretreatment with the different CPC formulations tested. Bars represent the percentage of GFP-expressing cells, compared to the infected condition, measured by flow cytometry. All determinations were performed in quadruplicate. Bars represent mean +/− SD of 4 independent determinations. Yellow bars, positive controls containing SDS or CPC; Blue bars, PerioAid-based mouthwashes; Green bars, Vitis-based mouthwashes. Un, uninfected; Inf, infected untreated; PerA++, PerioAid containing CPC and chlorhexidine; Vitis+; Vitis containing CPC.

**Table 1 viruses-15-01433-t001:** Composition of mouthwashes used in this study.

Formulation	Active Ingredient	
	Chlorhexidine	CPC
VITIS CPC Protect ^1^	-	0.07%
VITIS vehicle	-	-
PerioAid Intensive Care ^1^	0.12%	0.05%
PerioAid vehicle	-	-
PerioAid CHX	0.12%	-
PerioAid CPC	-	0.05%
CPC	-	0.05%

^1^ Commercial formula.

## Data Availability

Data supporting this study are included within the article.

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
