# Peer review of "Cetylpyridinium Chloride-Containing Mouthwashes Show Virucidal Activity against Herpes Simplex Virus Type 1"

_viruses, 2023, doi:10.3390/v15071433_

Round 1

Reviewer 1 Report

1) The main protocol for testing antiviral efficacy is not provided. Details of viral inhibition experiments are insufficient. The protocol for virus treatment and clean up must be described with sufficient detail to be reproduced. (how many infectious HSV or HPV particles were used based on titration? which dilutions of mouthwash were tested? Experimental volumes, times, and concentrations must be provided.

2) The composition of mouthwash vehicles should be included.  This reviewer assumes it would have been in Table1, which is mentioned (line 103) but not provided.

3) Line 29: there are only two types of HSV (HSV-1 and HSV-2). Currently there are nine recognized human herpesviruses. Although oral manifestations are associated with several HHVs, the oral cavity is not the main target for most of these infections (except for HSV-1).

4) Protocol for the generation of HPV pseudovirus must be referenced or described.

5)Title: needs to be modified. Only one  herpes virus was tested (HSV-1). The title could end in “against herpes simplex virus type 1.”

Minor points

Line 26: for accuracy, replace “members of the herpes simplex virus (HSVs) and human papillomavirus (HPV) families” with “members of the herpesvirus and human papillomavirus (HPV) families”

Line 29: replace “casing” by the common terminology “envelope”

Line 33: reference 9 is outdated, it only indicates that more that 60 HSV strains were known (unlike the more accurate number cited in the text).

Line 33: replace “an insurgence” with “the appearance”

Line 53: remove “by”

Line 93: references 26-28 are not adequate for this statement. They refer to tropism of HSV. They do not address the reported effects of CPC  in reducing viral load. Reference 30 is more appropriate.

Figure 1 does not have a sufficient resolution for reading.

Figure 2: abbreviations (Un Inf, perA++, Vitis+) must be defined.

Figure 2A: the x axis label should indicate what is measured by the instrument. Live&Dead is not sufficiently clear.

Figure 2BDF: explain the color-coding of bars in these graphs.

Figure 2D: For clarity and comparison, figure 2D and 2F should represent data in the same way. Figure 2D should also show the relative percentages compared to untreated sample (100% infection), similarly to panel 2F, which is a better representation of blockade of infection.

Line 109: reference 26 does not address the claim of this sentence.

Line 113: Regarding cell viability testing, “specific dye” is an insufficient description. Refer to the testing kit and indicate what is measured by flow cytometry.

Line 115: (mean SD, p value) should be (mean, SD, p-value). p-value should be indicated in the graph so support the claim of a significant difference compared to control. Also provide p value for PerioAid++ to show the viability difference is not significant.

Line 125: reference 29, address SDS inactivation of HIV. Also provide original references for HSV and HPV since those are most relevant for this study.

Throughout the manuscript, make sure that decimal numbers use periods and not commas.

No major issues.

Author Response

1) The main protocol for testing antiviral efficacy is not provided. Details of viral inhibition experiments are insufficient. The protocol for virus treatment and clean up must be described with sufficient detail to be reproduced. (how many infectious HSV or HPV particles were used based on titration? which dilutions of mouthwash were tested? Experimental volumes, times, and concentrations must be provided.

Following the reviewer suggestion, we have included a detailed description of the protocols used in the material and method section, including infectious particles used, clean up protocol and experimental volumes (lines 91-97).

2) The composition of mouthwash vehicles should be included.  This reviewer assumes it would have been in Table1, which is mentioned (line 103) but not provided.

We greatly appreciate the reviewer’s remark in the composition of the mouthwashes, since they are a key part of the effectiveness of their formulations. VITIS CPC Protect and PerioAid Intensive Care mouthwashes are commercially available. Specifically, the composition of the vehicles used in this study is: Deionized Water, Humectant, Sweetener, Emulsifier, Aroma and Preservatives. None of them possesses antiviral or antibacterial activity by itself. We have now included the composition of the distinct active principles of the mouthwashes used in Table 1. The composition of the vehicles is also described in the materials and methods section (lines 60-63).

3) Line 29: there are only two types of HSV (HSV-1 and HSV-2). Currently there are nine recognized human herpesviruses. Although oral manifestations are associated with several HHVs, the oral cavity is not the main target for most of these infections (except for HSV-1).

 We have corrected the description of herpesviruses provided in the introduction to clarify the number and tropism of human herpesviruses (lines 29-31). Indeed, we use HSV-1 for in vitro tests, due to the reported capacity of infection oral cavity. 

4) Protocol for the generation of HPV pseudovirus must be referenced or described. 

The protocol for the generation of HPV pseudovirus is now included in the materials and methods section (lines 80-86). 

5)Title: needs to be modified. Only one herpes virus was tested (HSV-1). The title could end in “against herpes simplex virus type 1.”

 We have changed the title accordingly.

Minor points 

Line 26: for accuracy, replace “members of the herpes simplex virus (HSVs) and human papillomavirus (HPV) families” with “members of the herpesvirus and human papillomavirus (HPV) families

The sentence has been corrected.

Line 29: replace “casing” by the common terminology “envelope”

We have replaced the term “casing” by “envelope”, as suggested by the reviewer.

Line 33: reference 9 is outdated, it only indicates that more that 60 HSV strains were known (unlike the more accurate number cited in the text).

Reference 9 referring to HPV subtypes has been updated.

Line 33: replace “an insurgence” with “the appearance”

Replaced.

Line 53: remove “by”

Corrected.

Line 93: references 26-28 are not adequate for this statement. They refer to tropism of HSV. They do not address the reported effects of CPC  in reducing viral load. Reference 30 is more appropriate.

We have changed references 26-28 for reference 30.

Figure 1 does not have a sufficient resolution for reading.

Thanks for the comment, we have improved resolution of figure 1.

Figure 2: abbreviations (Un Inf, perA++, Vitis+) must be defined.

Abbreviations are now described in the figure legend. Vitis is not abbreviation as it is a registered trademark from Dentaid SL.

Figure 2A: the x axis label should indicate what is measured by the instrument. Live&Dead is not sufficiently clear.

Following the reviewer suggestion, x axis now indicates the dye of the Live&Dead kit used, in this case APC-Cy7.

Figure 2BDF: explain the color-coding of bars in these graphs.

Colour-coding is now better described in the figure legend.

 Figure 2D: For clarity and comparison, figure 2D and 2F should represent data in the same way. Figure 2D should also show the relative percentages compared to untreated sample (100% infection), similarly to panel 2F, which is a better representation of blockade of infection.

We agree with the reviewer and thus, we have represented figure 2D in relative percentage compared to untreated sample. However, the new figure shows little difference with previous one as, unlike HPV infection, HSV virus is very efficient and nearly all cells are infected in the infected control, as seen also in the flow cytometry example from figure 2D, upper right panel.

 Line 109: reference 26 does not address the claim of this sentence.

We have replaced reference 26 by a more appropriate one.

 Line 113: Regarding cell viability testing, “specific dye” is an insufficient description. Refer to the testing kit and indicate what is measured by flow cytometry.

As stated above, we have included the exact reference of the Live&Dead kit used, including the dye, APC-Cy7.

 Line 115: (mean SD, p value) should be (mean, SD, p-value). p-value should be indicated in the graph so support the claim of a significant difference compared to control. Also provide p value for PerioAid++ to show the viability difference is not significant.

Following the reviewer suggestion we have included the p-values in the corresponding results section (lines 134-135).

Line 125: reference 29, address SDS inactivation of HIV. Also provide original references for HSV and HPV since those are most relevant for this study.

We have included more appropriate references demonstrating HSV and HPV inactivation by SDS and deleted reference 29.

Throughout the manuscript, make sure that decimal numbers use periods and not commas.

We have checked all the manuscript and changed periods for commas in decimal numbers.

Reviewer 2 Report

In this study the authors provide evidence that mouthwashes containing Cetylpyridinium chloride (CPC) exhibits significant antiviral activity against enveloped viruses, such as HSV, but not against non-enveloped viruses, such as HPV. 

The results presented in this study indicate that CPC -containing mouthwashes can serve as a safe and cost-effective measure to limit the infection and transmission of enveloped viruses in the oral cavity, contributing to the reduction of viral spread. However, I have some concerns regarding the potential toxicity of CPC and the possible side effects associated with long-term use of CPC-containing mouthwashes. 

The authors showed that short exposures (2 minutes) of Vero cells to CPC formulations did not cause a significant reduction in cells viability. However, the study lacks reported results regarding cell toxicity in 293TT, the other cell line used in this investigation.  

CPC is quaternary ammonium compounds with positive charges that exhibit a strong affinity for the phospholipid bilayer of the plasma membrane. Consequently, the long alkyl chains associated with CPC perturb the bilayers, leading to damage in the plasma membrane. The authors mentioned briefly in the manuscript the ability of CPC to damage plasma membrane but there are other problems associated to CPC. In fact, inhalation of aerosolized QACs has been associated with adverse effects on the respiratory system, causing pulmonary toxicity and inflammation in both humans and animal models. Additionally, CPC has been shown to alter membrane surface tension and fluidity, induce apoptosis in alveolar epithelial cells, and promote pulmonary epithelial mesenchymal transition.  

It could be argued that the mentioned side effects are concentration-dependent; however, the authors need to address this concern convincingly and provide evidence to support their claims. 

Quality of the English was acceptable. 

Author Response

In this study the authors provide evidence that mouthwashes containing Cetylpyridinium chloride (CPC) exhibits significant antiviral activity against enveloped viruses, such as HSV, but not against non-enveloped viruses, such as HPV. The results presented in this study indicate that CPC -containing mouthwashes can serve as a safe and cost-effective measure to limit the infection and transmission of enveloped viruses in the oral cavity, contributing to the reduction of viral spread. However, I have some concerns regarding the potential toxicity of CPC and the possible side effects associated with long-term use of CPC-containing mouthwashes. 

The authors showed that short exposures (2 minutes) of Vero cells to CPC formulations did not cause a significant reduction in cells viability. However, the study lacks reported results regarding cell toxicity in 293TT, the other cell line used in this investigation.  

The antiviral activity was performed only in Vero cells. 293TT cells were only used for virus production. To clarify this issue, we have now included a more detailed explanation of the protocols used both for antiviral testing and for virus production in the corresponding materials and methods section (lines 80-86 and 91-97).

CPC is quaternary ammonium compounds with positive charges that exhibit a strong affinity for the phospholipid bilayer of the plasma membrane. Consequently, the long alkyl chains associated with CPC perturb the bilayers, leading to damage in the plasma membrane. The authors mentioned briefly in the manuscript the ability of CPC to damage plasma membrane but there are other problems associated to CPC. In fact, inhalation of aerosolized QACs has been associated with adverse effects on the respiratory system, causing pulmonary toxicity and inflammation in both humans and animal models. Additionally, CPC has been shown to alter membrane surface tension and fluidity, induce apoptosis in alveolar epithelial cells, and promote pulmonary epithelial mesenchymal transition.   

The authors fully understand the concern about the use of a molecule like CPC and its possible toxicity. As the reviewer comments, it is a molecule with affinity for the membrane and capable of disturbing it, in fact, it is an antiseptic widely used during the last decades. Concerns about the toxicity caused by the molecule when inhaled as aerosol form are perfectly reasonable and understandable, but its use as an active ingredient in commercial mouthwashes has been proven safe for decades. First of all, it is an oral hygiene product that is not ingested but expelled after successive gargles. Finally, it is a compound accepted by both the FDA (UNII D9OM4SK49P) and the Scientific Committee on Consumer Safety of European Union (SCCS/1548/15) for its use in gingivitis as a mouthwash as long as CPC concentration does not exceed 0.1% (w/w). Indeed, our in vitro experimental protocol is designed to mimic the procedure of a mouthwash, i. e., short-time exposure (1 or 2 min) followed by extensive washing. To clarify these issues, we have included a more detailed description on CPC use and potential toxicity issues and the protocol used for in vitro determining antiviral activity (lines 127-128 and 91-97).

It could be argued that the mentioned side effects are concentration-dependent; however, the authors need to address this concern convincingly and provide evidence to support their claims. 

As stated above, we have not tested CPC activity in a dose-dependent manner as the objective of the work was to determine the antivirucidal potential of the commercialized formula of the corresponding mouthwashes. We have clarified this issue in the abstract and introduction sections, when describing the aim of the study (lines 54-57). In addition, we have included a sentence in the discussion, highlighting the use of approved formula (lines 137-141).  

Round 2

Reviewer 1 Report

The authors adequately responded to previous comments and concerns.